# Alpha-Emitting Radionuclides: Current Status and Future Perspectives

**DOI:** 10.3390/ph17010076

**Published:** 2024-01-08

**Authors:** Matthias Miederer, Martina Benešová-Schäfer, Constantin Mamat, David Kästner, Marc Pretze, Enrico Michler, Claudia Brogsitter, Jörg Kotzerke, Klaus Kopka, David A. Scheinberg, Michael R. McDevitt

**Affiliations:** 1Department of Translational Imaging in Oncology, National Center for Tumor Diseases (NCT/UCC), 01307 Dresden, Germany; 2Medizinische Fakultät and University Hospital Carl Gustav Carus, Technische Universität Dresden, 01307 Dresden, Germany; 3German Cancer Research Center (DKFZ), 69120 Heidelberg, Germany; 4Helmholtz-Zentrum Dresden-Rossendorf (HZDR), 01328 Dresden, Germany; 5Research Group Molecular Biology of Systemic Radiotherapy, German Cancer Research Center (DKFZ), 69120 Heidelberg, Germany; m.benesova@dkfz-heidelberg.de; 6Helmholtz-Zentrum Dresden-Rossendorf, Institute of Radiopharmaceutical Cancer Research, Bautzner Landstr, 400, 01328 Dresden, Germany; 7School of Science, Faculty of Chemistry and Food Chemistry, Technische Universität Dresden, 01062 Dresden, Germany; 8Department of Nuclear Medicine, University Hospital Carl Gustav Carus, Technische Universität Dresden, Fetscherstraße 74, 01307 Dresden, Germany; david.kaestner@ukdd.de (D.K.); claudia.brogsitter@ukdd.de (C.B.); 9National Center for Tumor Diseases (NCT) Dresden, University Hospital Carl Gustav Carus, Fetscherstraße 74, 01307 Dresden, Germany; 10German Cancer Consortium (DKTK), Partner Site Dresden, Fetscherstraße 74, 01307 Dresden, Germany; 11Molecular Pharmacology Program, Sloan Kettering Institute, New York, NY 10065, USA; scheinbd@mskcc.org; 12Molecular Imaging and Therapy Service, Department of Radiology, Memorial Sloan Kettering Cancer Center, New York, NY 10065, USA; 13Department of Radiology, Weill Cornell Medical College, New York, NY 10065, USA

**Keywords:** alpha emitter, targeted alpha therapy, actinium-225, high let, theranostic

## Abstract

The use of radionuclides for targeted endoradiotherapy is a rapidly growing field in oncology. In particular, the focus on the biological effects of different radiation qualities is an important factor in understanding and implementing new therapies. Together with the combined approach of imaging and therapy, therapeutic nuclear medicine has recently made great progress. A particular area of research is the use of alpha-emitting radionuclides, which have unique physical properties associated with outstanding advantages, e.g., for single tumor cell targeting. Here, recent results and open questions regarding the production of alpha-emitting isotopes as well as their chemical combination with carrier molecules and clinical experience from compassionate use reports and clinical trials are discussed.

## 1. Introduction

The ensemble between diagnostic and therapeutic radionuclides has opened a rapidly growing area for individualized targeted radionuclide theranostics. Recent developments and increasing knowledge in the field of radiation biology, radiochemistry, radiopharmaceutical sciences, nuclear medicine, and oncology are currently making steps forward to several more, highly relevant therapy options. One distinct research field is that of the therapeutic use of alpha particle-emitting radionuclides (alpha emitters). With their unique physical properties, it has been hypothesized that principles known, e.g., from external proton irradiation can be transferred to a systemic targeted internal radiotherapy, also called targeted endoradiotherapy, down to treating single cells. The physical properties of alpha emitters determining their biological potential are their short range in tissue with 50–80 µm and their high linear energy transfer (LET) along this track (Figure 1). With typical particle energies of 5–9 MeV, the resulting LET is approximately 80–100 keV/µm, which is orders of magnitude higher compared to beta or gamma radiation.

The main target of radiation is the cell nucleus and when comparing alpha to beta particles, 2 to 10-fold higher relative biological effectiveness can be measured [1,2]. High LET radiation results in both extremely high radiotoxicity per alpha particle and thus in a cytotoxic effect that is at least partly independent of the formation of reactive oxygen species. This has been known to be advantageous, particularly for the treatment of hypoxic tumors. The alpha particle-induced DNA damage often leads to complex double-strand breaks (DSBs). These DSBs are the assumed mechanism that drives an exponential curve of dose–response to alpha radiation. This is in contrast to beta or gamma irradiation, which induces cell death with a dose–response relationship described by a linear quadratic model. The high LET also leads to the strongly reduced dependency of a damaging effect on oxygenation. Thus, for hypoxic and radio-resistant cells, alpha radiation has further advantages as a targeted therapeutic agent over other forms of radiation.

For clinical development and application, the theranostic pairing of diagnostic and therapeutic isotopes has posed certain limitations when applied to the use of therapeutic alpha emitters. Diagnostic radionuclides can be utilized to characterize key pharmacokinetic parameters such as tumor uptake, clearance, and any accumulation in off-target organs and tissues. Furthermore, these diagnostic radionuclides can yield additional biological information about the tumor target and dosimetry predictions of therapeutic radionuclides. However, dosimetry predictions for internal targeted alpha therapy (TAT) have limitations. This is due to possible additional factors that are partly unique for alpha radiation. One is that the heterogeneity of activity distribution on a microscopic level has a much higher impact on efficacy, since the range of alpha particles is limited to a few cell diameters. Another obstacle preventing the accurate prediction of effects from imaging is the common use of alpha-emitting isotopes derived from in vivo isotope generators. That is, after the decay of the original targeted isotope, additional alpha emitting as well as beta emitting daughter nuclides are released which contribute to cytotoxic effects. Typically, these daughter radionuclides have independent and often difficult to model pharmacokinetic profiles due to local protein or cell binding and heterogeneous clearance [3].

Here, we review current strategies on ways that alpha particle-emitting radionuclides (Figure 2) can be provided on recent developments of the chemistry to radiolabel them to molecules functioning as binding vectors including the new hypothesis of matched radionuclide pairs that might foster clinical development and on the currently available clinical experience. 

## 2. Current Production and Availability of Alpha-Particle-Emitting Radionuclides

Actinium-225 (^225^Ac) could be seen as the most critical radionuclide because of its high demand and limited availability. The primary source of ^225^Ac comes from uranium-233 (^233^U, τ½ = 1.59…10^5^ y) waste which was generated in the frame of nuclear weapons development 80 years ago. ^233^U decays to thorium-229 (^229^Th; τ½ = 7.88…10^3^ y), which serves as a parent radionuclide for radionuclide generators and provides a robust source of ^225^Ac each month [4].

Unfortunately, the amount of available ^233^U/^229^Th is sufficient to generate ^225^Ac in MBq/mCi quantities, enabling less than a thousand of targeted alpha therapies per year [5]. This issue comes hand in hand with not only uncertain ^225^Ac availability, but also higher costs and limited research options which hamper the widespread translation and application of targeted alpha therapies. Thus, solving this shortage by alternative production routes became one of the main priorities in the community involving both scientists and physicians. The current focus lies on the cyclotron and reactor production with accompanying purification strategies while employing existing infrastructure and approaches as well as securing other infrastructural and technical alternatives. One concrete example might be demonstrated by the joint effort of Brookhaven, Los Alamos, and Oak Ridge National Laboratories to deliver accelerator-produced ^225^Ac [6].

There are various nuclear reactions either producing ^225^Ac directly or providing the parent radionuclide thorium-232, thorium-229 or radium-225, alternatively. These include but are not limited to: ^226^Ra(γ,,n)^225^Ra → ^225^Ac (accelerator, electrons); ^226^Ra(p,2n)^225^Ac, ^226^Ra(α,n)^229^Th, ^232^Th(p,x)^229^Th, ^226^Ra(p,pn)^225^Ra, (accelerator, low-energy particles); ^232^Th(p,x)^225^Ac, ^232^Th(p,x)^225^Ra → ^225^Ac (accelerator, high-energy particles); and ^226^Ra(3n, γ)^229^Ra → ^229^Ac → ^229^Th (reactor, thermal neutrons) [7,8,9].

The main complication of ^225^Ac production via the accelerator applying higher energies results in the co-production of actinium-224 (τ½ = 2.78 h), actinium-226 (τ½ = 29.37 h), and actinium-227 (τ½ = 21.77 y)—which cannot be separated from the desired actinium-225. Since the half-lives of actinium-224 and -226 are very short, both are eliminated by natural decay. On the other hand, the long half-life of actinium-227 significantly complicates its related waste disposal, dosimetry, and radiation safety [10]. The actinium-225/actinium-227 ratio improves with increasing proton energy, but degrades with a longer irradiation time and careful balance between these two parameters has to be set. In addition, the irradiation of a highly radiotoxic and not so easily accessible radium-226 (τ½ = 1.600 y) in a cyclotron needs a more sophisticated target with gas-trapping filters and/or leak-tight equipment due to the presence of radioactive noble gas radon-222 (τ½ = 3.82 d) [11]. It is also important to note that the accelerator-produced ^225^Ac differs slightly in comparison to the one obtained from the radionuclide generator, which, e.g., requires separate drug master files.

Another possibility is also to focus on the development, evaluation, and application of another alpha in vivo nanogenerator, thorium-227 (^227^Th, τ½ = 18.9 d) [12]. There is, however, one significant difference between ^227^Th and ^225^Ac, which is that they have concretely different equilibria. ^225^Ac forms a so-called secular equilibrium (τ½, parent >>> τ½, decay nuclide) with its decay radionuclide francium-221 having the half-life of 5 min. ^227^Th forms a so-called transient equilibrium (τ½, parent > τ½, decay nuclide) with its decay radionuclide radium-223 having the half-life 11.4 d. By the direct comparison of the 5 min half-life of francium-221 with the 11.4 d half-life of 11.4 d results in the fact that ^227^Th does not deliver so many alpha particles as quickly as ^225^Ac and, by that, it has a much higher radiobiological effectiveness, which is clearly desired for targeted alpha therapies [13].

## 3. Radiochemistry and Concept of Theranostic Matched Radionuclide Pairs

A fundamental requirement of using alpha emitters for treatment is their selective delivery in vivo to a cancer cell target. This is ideally pursued with molecules used as binding vectors that display high target accumulation and low interactions with off-target sites. Also, it is essential that the sufficient stability of the labeling is ensured along with minimal influence on the initial pharmacokinetic properties of the carrier. By matched radionuclide pairs, theranostics allow the combination of diagnostic imaging using, for example, positron-emitting radionuclides with the therapeutic approach using particle-emitting radionuclides. Matched radionuclide pairs are generally a crucial component of radionuclide theranostics since the pharmacokinetics of the binding vector and the availability of the tumor targets are the main factors that determine the efficacy of radioligand-mediated endoradiotherapy. Theranostic pairs consist of the same binding vector molecule and matching radionuclides that share in best case identical or at least similar chemical properties, making them ready to label one and the same precursor compound. Therefore, the exact prediction of the therapeutic radionuclide distribution can be derived from the more suited imaging. 

The key criteria for the choice of alpha-particle-emitting radionuclides are the half-life, the stable binding to a chelating system, the particle energy, the possible decay chain properties, and the kinetics of the daughters, as well as the costs and availability. These features result in a small number of radionuclides that are suitable, including thorium-227, actinium-225, radium-223/-224, bismuth-212/-213, astatine-211, and terbium-149 [14,15]. Additionally, lead-212 can be mentioned in this list, while delivering one beta particle prior to the alpha decay. Some attempts were made to use the concomitant gamma radiation of the alpha radionuclides as the diagnostic modality, but this has been met with limited success in human applications, mostly due to the activities used for alpha radionuclide therapy that are orders of magnitude lower than they are for imaging. To fulfill the theranostic concept, the respective matching radionuclides for imaging were previously developed using the same conjugate or precursor compound.

### 3.1. Matched Radionuclide Pair Lanthanum-133/Actinium-225

All isotopes of actinium are radioactive. Among them, ^225^Ac has favorable nuclear properties such as a half-life of 9.9 days and a decay chain delivering 4 alpha and 2 beta particles [16]. Lanthanum has similar coordination properties and can be used as a nonradioactive match for the design of new radioconjugates and as a diagnostic reference isotope. Recently, two ß^+^ emitters were introduced with lanthanum-132 (^132^La) and lanthanum-133 (^133^La) as theranostic matches [17].

Actinium mainly exists as a cation in the oxidation state +3. Different chelating compounds were developed in the past, especially for radiopharmaceutical applications. As a hard cation according to the hard–soft acid–base (HSAB) concept, the Ac^3+^ cation prefers oxygen donor atoms for complexation in coordination numbers of >10, but oxygen and nitrogen mixed ligands were mostly applied. The most sufficient complex stability is found with open-chain ligands of high denticity like DTPA which is increased when changing to macrocyclic ligands like DOTA [16] (Figure 3). 

The first attempts to determine the in vivo behavior of ^225^Ac complexes were made using citrate, DTPA, and EDTMP as ligands [18]. A rapid radiolabeling kinetic is found with these acyclic ligands, allowing a fast complexation of the radiometal at ambient temperatures within minutes. However, these formed complexes are kinetically labile, leading to a release of the radiometal in vivo. As a result, significant amounts of ^225^Ac as well as of its (grand)daughters were found in the liver and the femur [19,20]. A higher stability can be reached when using ligands with a higher donor number. For example, the more stable 2^25^Ac complex is formed using CHX-A″–DTPA with eight donor functions in contrast to EDTA with six donor positions. Additionally, the steric effect and the certain pre-organization of the backbone of the CHX-A″–DTPA ligand have a positive effect on the complex stability [19].

The aforementioned pre-organization of the chelating system is always found in macrocyclic compounds and is known as a macrocyclic effect, mostly leading to the higher stability of the complexes [21]. In this regard, macrocycles such as HEHA, PEPA, TETA, TETPA, and DOTPA were employed for ^225^Ac complexation. These all differ in cavity size and donor numbers, but the stability of the formed ^225^Ac complexes may still be lacking. For instance, the ^225^Ac–HEHA complex is more stable in vivo compared to DOTA due to the higher donor number of 12 and the larger cavity for the metal ion. In contrast, the functionalized HEHA–NCS, which was used for antibody labeling with ^225^Ac, has a low in vivo stability and decomposed to 50% after 24 h when tested in fetal bovine serum [22].

Due to the convenient availability, most experience is cumulated with the macrocyclic chelator DOTA (1,4,7,10-tetraazacyclododecane-*N*,*N*′,*N*″,*N*′′′-tetraacetic acid) used in a series of radiopharmaceuticals suitable for clinical use [23]. DOTA is a 12-membered macrocycle containing four tertiary amine nitrogen donors and four pendent arms with carboxylate functional groups that altogether provide an octadentate coordination. Due to the ability to stably bind the hard cations of charge +3, DOTA is expected to work well as a chelating agent for Ac^3+^ [24].

The DOTA chelator [25] has been widely used for the alpha-emitting radionuclides actinium-225, bismuth-213, and terbium-149, frequently for thorium-227, and some trials were made for radium-223 [12,26,27,28]. Elevated temperatures of up to 90–100 °C and a labeling time of 15–30 min were typically required for labeling, making this chelator unfavorable for sensitive biomacromolecules like proteins or antibodies. For this purpose, a robust clinical labeling method was developed, consisting of a two-step labeling strategy (pre-labeling) [29].

A variety of trivalent radiometal cations for diagnostic applications like ^111^In, ^43/44^Sc, or ^68^Ga were also stably complexed with DOTA, forming matched pairs with ^225^Ac and even with the therapeutic beta emitter ^177^Lu, but also mostly under the same unfavorable labeling conditions [25]. Several EMA- and FDA-approved radioconjugates, such as [^68^Ga]Ga-DOTA-TATE, [^177^Lu]Lu-DOTA-TATE, or [^177^Lu]Lu-PSMA-617, are known to contain DOTA as a chelating motif. 

The sufficient in vivo stability was pointed out in a preliminary biodistribution study of [^225^Ac]Ac–DOTA in normal BALB/c mice only showing a slight accumulation in the liver (3.29% ID/g) and bone (2.87% ID/g) after 5 days [20]. The high efficiency of the ^225^Ac chelation was further demonstrated using DOTA–NCS in a two-step labeling procedure to create ^225^Ac–DOTA-modified IgG antibodies to avoid denaturation during radiolabeling [29]. In the first step, ^225^Ac–DOTA–NCS was prepared from 2B–DOTA–NCS at 55–50 °C within 30 min. In the second step, ^225^Ac–DOTA–NCS was conjugated to the antibody at 37 °C for 52 min via a free lysine function. Notably, the ^225^Ac–DOTA complex showed a slow dissociation with a loss of 10% over one half-life. Several antibodies such as HuM195 (antiCD33), B4 (anti-CD19), trastuzumab (anti-HER2/neu), and J591 (anti-PSMA), which target leukemia, lymphoma, breast/ovarian cancer, and prostate cancer, respectively, were ^225^Ac-labeled using this procedure and tested in vivo [30]. One recent innovation is the development of a live-cell-based theranostic carrier, in which a chimeric antigen receptor (CAR) T cell can chelate the PET diagnostic and ^225^Ac therapeutic isotopes for delivery to the tumor cell and live tracking in vivo [31,32,33]. These promising initial results triggered a wave of investigations that resulted in numerous clinical trials [34].

### 3.2. Chelator Design to Improve the In Vivo Stability of ^225^Ac Complexes

In 2017, a new chelating macrocycle called macropa (*N*,*N′*-bis[(6-carboxy-2-pyridyl)methyl]-4,13-diaza-18-crown-6) was introduced, allowing radiolabeling within a 5 min labeling time at room temperature with >99% RCC and leading to remarkably stable in vivo ^225^Ac–macropa complexes [35] (Table 1). Subsequently, an NCS-modified and two clickable derivatives were prepared, allowing the conjugation to target vector molecules [36,37].

In vitro and in vivo studies showed the remarkable stability of the ^225^Ac-mcp-radioconjugates over a period of 10 days and a high tumor accumulation combined with a fast renal excretion in LNCaP-tumor bearing mice. In 2022, a ^225^Ac-radioconjugate based on the anti-EGFR antibody ch806 was presented, showing high stability when challenged with La^3+^ and EDTA in human serum. High tumor accumulation was found in U87MG.de2–7 xenografts. A therapy study showed 100% survival of the tumor-bearing treatment group over 80 days post-injection [38]. Recently, [^225^Ac]Ac–MACROPATATE, the macropa-variant of DOTATATE based on the Tyr^3^-octreotate peptide, was developed to treat neuroendocrine tumors [39]. The remarkable radiotracer stability of 10 days in human serum was confirmed and a high accumulation in SSTR-positive tumors was pointed out in mice bearing SSTR-positive H69 tumor xenografts. However, a higher off-target accumulation of [^225^Ac]Ac–MACROPATATE was found.

[^225^Ac]Ac–crown–αMSH was developed containing a chelator with a tetraazacrown-6 backbone with four pendant acetate side arms, which is connected to a peptide to target the melanocortin 1 receptor (MC1R) in specifically expressed primary and metastatic melanoma [40]. The radiolabeling worked under mild conditions (pH 5–7, rt, 10 min, c = 10^−7^ M, >98% RCY). However, the in vivo stability of the resulting tracer was lacking, which was indicated from the time-dependent HPLC experiments over 16 h. A different biodistribution profile was obtained when using freshly prepared the ^225^Ac-crown-αMSH in contrast to the overnight prepared sample, showing the insufficient in vitro stability of this chelator.

Furthermore, BZmacropa–NCS was developed containing a benzyl moiety in the macrocyclic ring to investigate this modification on the complexation stability. A respective antibody conjugate GC33-BZmacropa was preclinically investigated showing a slightly reduced in vivo stability and higher uptake in the liver and femur [41].

One drawback of the macropa-based chelators is the absence of diagnostic radionuclides, because the standard radionuclides used for DOTA failed here. Thus, the radioisotopes of lanthanum, namely ^132^La (τ½ = 4.6 h, E_β+,mean_ = 1.29 MeV) and ^133^La (τ½ = 3.9 h, E_β+,mean_ = 0.46 MeV) were utilized as β^+^ emitters for PET [17]. Lanthanum has a similar coordination chemistry to actinium and therefore acts as a surrogate (ionic radii: 1.12 Å for Ac^3+^ and 1.032 Å for La^3+^ in six-fold coordination). Highly apparent molar activities up to 330 GBq/µmol and a high in vivo stability were observed for ^133^La–macropa complexes using a macropa concentration down to 10^−7^ M [42,43,44]. PET phantom images were performed, pointing out that the spatial resolution and contrast of ^133^La is superior to those of ^44^Sc, ^68^Ga, and ^132^La, but comparable to ^89^Zr (E_β+,mean_ = 0.396 MeV) [45].

^13X^La-radioconjugates, namely [^133^La]La–PSMA–I&T, [^133^La]La–macropa–DUPA, [^133^La]La–DO2APic–DUPA, [^132^La]La–NM600 and [^133^La]La–mcp–M–PSMA were prepared on the basis of DOTA, DO3APic, and macropa as chelators [42,43,45,46].

## 4. Radioisotopes of Lead for Theranostics

Radioisotopes of lead, ^212^Pb (β emitter, τ½ = 10.6 h), and ^203^Pb (γ emitter, τ½ = 51.9 h) as the true matched pair have gained attention for TAT, because ^212^Pb functions as an in vivo generator for the release of ^212^Bi (τ½ = 1.01 h), which is the actual α emitter [47,48]. ^212^Pb can be obtained as the decay nuclide from ^224^Ra. Different chromatographic generator systems were developed to isolate ^212^Pb based either on ^228^Th as the mother nuclide or directly on ^224^Ra [49]. ^203^Pb is available via cyclotron irradiation of enriched ^203^Tl [49,50]. As a borderline cation according to the hard–soft acid–base concept, Pb^2+^ is found in complexes with oxygen, nitrogen, and sulfur donor atoms and coordination numbers between 2 and 10 [51]. In aqueous environments, Pb is primarily found as a bivalent cation in the oxidation state +2. The most frequently used chelators for radioconjugate preparation are DOTA [52] and TCMC or DOTAM (1,4,7,10-tetraaza-1,4,7,10-tetra(2-carbamoylmethyl)cyclododecane) and the derivatives thereof, whereas DOTAM seems to have a higher in vivo stability compared to DOTA [14]. A total CN of 8 was found in the Pb complexes of the standard chelators DOTA, DOTAM, or DTPA with high log K values of >18. In addition to the true match ^203^Pb, other standard diagnostic radionuclides like ^43/44^Sc, ^68^Ga, or ^111^In can possibly be used for diagnostic purposes depending on the chelating system. 

Recently, ^203/212^Pb came into focus by several first in-human theranostic applications. In 2014, [^212^Pb]Pb–TCMC–trastuzumab was used for patients with human epidermal growth factor receptor type 2 (HER-2)-expressing malignancies [53]. The antibody was modified with DOTAM as a chelator with preference over DOTA. The in vitro and animal model testing of [^212^Pb]Pb–TCMC–trastuzumab to investigate the therapeutic behavior prior to the human trials was performed to give an explicit preference of DOTAM as the chelator over DOTA [54].

To treat metastatic SSTR-expressing neuroendocrine tumors, the radioconjugate [^212^Pb]Pb–DOTAMTATE was used in a first-in-human dose-escalation clinical trial with 10 patients [55]. However, the diagnostic imaging was performed with [^68^Ga]Ga–DOTATATE, whereas [^203^Pb]Pb–DOTAMTATE was used for human dose calculations. The activity dose was administered in four circles leading to an effective reduction in tumor lesions. Additionally, the Tyr^3^–octreotide (TOC) variant VMT-α-NET was used for human applications as a conjugate for ^203/212^Pb that shows high chelation properties [56]. SPECT/CT images (low dose, 224 MBq) using [^203^Pb]Pb–VMT-α-NET were acquired to assess the feasibility of the [^212^Pb]Pb–VMT-α-NET therapy. A higher NET uptake combined with a rapid renal excretion within the first hour was observed.

Ligands with the PSMA-617-binding motif containing the chelators p-SCN–Bn–DOTAM or DO3AM were used for preclinical studies with ^203^Pb [57]. Interestingly, the slightly different coordination behaviors of the chelating ligands to Pb^2+^ resulted in a different tumor uptake and internalization in vitro. [^203^Pb]Pb–PSMA–CA012 was found to be the best candidate showing a high tumor uptake and internalization combined with a fast renal excretion. 

### Astatine-211: The Alpha-Emitting Therapeutic Big Brother of Radioiodine

Since the first discovery of ^211^At in 1940, several reports on human therapy treatments with ^211^At are known [58]. A 100% alpha emission with only one alpha particle emitted per decay was found for ^211^At which prevented the unpredictable dose localization caused by the formation for radioactive daughters. ^211^At is cyclotron-produced by the irradiation of ^209^Bi with α-particles accelerated at ∼28 MeV. Its chemistry resembles iodine; however, its covalent bonds are more instable. Furthermore, it also has a tendency to behave like a metalloid. Nonetheless, naturally occurring ^127^I is used as a nonradioactive reference and the radioisotopes of iodine like ^123^I or ^124^I function as diagnostic matches. Oxidation states from +7 to –1 are possible, but −1 oxidation state is probably the most clearly established form of astatine with strong similarity to iodide [59]. It can easily be converted into the +1 oxidation state using mild oxidizing agents, such as Chloramine-T, Iodogen, or *N*-halosuccinimide to generate electrophilic I^+^ or At^+^ to perform electrophilic reactions. In this regard, labeling strategies are related to the formation of covalent bonds with carbon in most the cases. However, the carbon–astatine bond is much weaker compared to the carbon–iodine bond, but with a higher stability of astatine–aryl compounds over astatine–alkyl compounds, as expressed in Table 2.

Several clinical trials were made in the past, with Na [^211^At]At itself (thyroid cancer) or small organic molecules like [^211^At]At-MABG (*meta*-[^211^At]astatobenzylguanidine) to treat malignant pheochromocytoma, but also with ^211^At-labeled biomacromolecules like proteins and antibodies [58]. In contrast to direct radiolabeling procedures with I^+^ using the tyrosine residues of the respective biomacromolecule, direct labeling with At^+^ is not possible. Differently labeled building blocks like *N*-succinimidyl 3-[^211^At]astatobenzoate ([^211^At]SAB) [60,61] were used in a two-step labeling approach. To further improve the labeling efficiency, one-step approaches were developed in which the radionuclide is directly reacted with a pre-conjugated biomacromolecule. *N*-succinimidyl 3-(trimethylstannyl)benzoate was first conjugated to the antibody (trastuzumab) and then labeled with ^211^At^+^ leading to a high RCY and A_S_ in a reduced procedure time. An improved approach using a cysteine coupling approach with an analogous maleimide-based precursor provides a more homogeneous bioconjugation to thiol instead of lysine residue [62]. Further improvements to raise the radioconjugate stability were made using guanidine-based building blocks like [^211^At]At-SAGMB [63] (Figure 4).

Recently, four new ^211^At-containing small-molecule radiotherapeutics based on the FAPI binding motif with different linkers (PEG, piperazine) were developed [64]. Cell uptake was performed using FAP-transfected HEK293/FAPα and A549/FAPα cell lines, and biodistribution on PANC-1-cell-bearing mice. Control experiments were performed with ^131^I-labeled derivatives. [^211^At]At-FAPI1 and [^211^At]At-FAPI5 were the most promising with the highest tumor uptake and the best therapeutic effect.

In order to limit uptake in thyroid or stomach tissues, alternative attempts were made towards the use of alternatives to astatine bound to carbon using three-dimensional carboranes [65], which form thermodynamically stable boron–astatine bonds. The bond enthalpy is estimated to be approximately 50% higher than the aryl–At bond [66]. This radiolabeling approach has been transferred in several successful preclinical therapy studies using monoclonal antibodies as carriers [58]. The frequently reported activity retention in the liver and kidney is a limit of these boron–At clusters, especially when using molecules that are smaller than monoclonal antibodies.

Alternatively, At–metal complexes became more prominent for astatine labeling due to the soft base character of the At–anion. The first investigations considered the formation of mercury complexes. Later, complexes with Rh^3+^ or Ir^3+^ as a central cation included in macrocyclic ligands were developed [67]. However, in vivo studies have not successfully proven these approaches adequate yet for further (pre)clinical applications [68]. Softer metal cations like Rh^+^ could improve the in vivo stability when *N*-heterocyclic carbenes are used as ligands [64,69].

## 5. Use of Carrier Molecules for Selective Delivery of Alpha Particle-Emitting Nuclides

The properties of alpha emitters are suitably matched to uses involving the targeting of various moieties ranging from small molecules over peptides to antibodies that are capable of selectively targeting receptors or antigens on cancer cells. A unique biological target that has been successfully addressed by targeted endoradiotherapy is the sodium–iodine symporter in thyroid cancer, which has been used for decades for radio-iodine therapy and can also be coopted for radioastatine-211 therapy since astatine chemically resembles iodine. When it comes to targeting in a broader sense, other synthetic binding vectors are introduced to facilitate target-specific transport in vivo (Figure 5). In nuclear medicine, several approaches for diagnostics and therapy are currently in clinical practice, namely the targets SSTR2, PSMA, CXCR4, αvβ3, αvβ6, and FAPα, among others [70,71]. Extensive experience of using pharmacokinetics with such small molecules has been documented in the literature and useful biological targets across several tumor entities are available. Based on the wide experience of measuring the uptake of these carriers by PET, robust data exist with regard to target availability that can be applied to targeted alpha therapies.

Other attractive carriers—taken from biological templates—are monoclonal antibodies. Theranostic approaches have also been suggested towards commonly known targets like Her2/neu [72]. The utilization of monoclonal antibodies as carriers for alpha-emitting radionuclides were widely used in the past and are promising for ongoing clinical trials [4,73]. Although monoclonal antibodies are associated with very good binding properties and a broad range of availability, their size limits renal elimination and therefore long circulation times are observed, which are associated with relevant toxicity. In contrast, small molecules can rapidly diffuse to targets leading to rapid accumulation together with fast excretion. This is then typically associated with a larger window between toxicity and efficacy for small molecules in contrast to monoclonal antibodies.

For example, the beta-emitting radionuclide lutetium-177 (^177^Lu) was used for labeling the PSMA-binding antibody J591, which displayed efficacy at a dose activity of two applications of 1.67 GBq/m^2^ myelosuppression as dose-limiting toxicity [74]. In contrast, ^177^Lu, which was used for labeling PSMA-617, was reported as safe in a phase 3 clinical trial when 4–6 cycles of 7.4 GBq were applied [75]. Small antibody-derived molecules such as nanobodies or antibody fragments might combine fast pharmacokinetics with retained high in vivo tumor binding. However, such approaches are still rarely applied in clinical trials. Also, the concept of separating tumor-binding pharmacokinetics from radionuclide-carrying pharmacokinetic pre-targeting approaches in order to reduce toxicity has rarely reached clinical trials to date.

## 6. Clinical Overview and Perspectives

Clinically, one commonly used alpha particle-emitting radionuclide is radium-223, with a randomized clinical trial in metastasized castration-resistant prostate cancer (mCRPC) showing a significant benefit in terms of overall survival compared to placebo, leading to the approval of radium-223 (^223^Ra, Xofigo^®^) in this indication. The approved activity is 55 kBq/kg, administered for six cycles every 4 weeks [76]. However, due to the indirect targeting mechanism aimed at bone remodeling and the insufficient ability to couple radium-223 with binding vector molecules, radium-223 has not been successfully used in other indications [77]. Nevertheless, the side effect profile of this treatment is low and its combination with targeted beta-emitting therapy is possible [78]. With the introduction of an alpha-emitting radionuclide into clinical routine, radiation safety has been addressed in particular. Possible doses that might arise from contamination with radium-223 were calculated. Also, theoretical estimations and measurements for clinical routines were performed. Although precautionary measures are dependent on the local regulatory authority, the application of radium-223 does not pose any significant radiation safety issue [79,80]. At the level of clinical trials or compassionate use programs, there is also firm experience for the use of many other alpha-emitters for several other indications.

In hematologic diseases, where tumor cells are readily accessible for carrier molecules and single cells are the main cause of tumor manifestations, the isotope bismuth-213 (τ½= 46 min) and the 9.9-day half-life radionuclide ^225^Ac were used to couple to the CD33-binding antibody lintuzumab (HuM195), and leukemia-inhibitory activity was reported [81,82,83]. In this case, the radioligand-constructs consisting of an alpha emitter and a monoclonal antibody were administered intravenously. The cellular target for lintuzumab is the myeloid-specific transmembrane receptor CD33, which has high availability on leukemia cells while lacking expression in hematologic stem cells. A maximum tolerated dose of 37 MBq/kg was reported for a sequential protocol with cytarabine followed by [^213^Bi]Bi–lintuzumab. The optimized targeting of lintuzumab was achieved by the administration of 250 μg unlabeled antibody before the administration of [^225^Ac]Ac-lintuzumab at activity levels between 18.4 and 148 kBq/kg. Based on a grade 3 hyperbilirubinemia and one episode of syncope complicated by subarachnoid hemorrhage toxicity, and both patients experiencing infections in the 148 kBq/kg dose group, the maximum tolerated dose was estimated at 111 kBq/kg as a single infusion of [^225^Ac]Ac–lintuzumab. With doses exceeding 37 kBq/kg, peripheral blasts were eliminated in most patients. However, in this small, severely therapy-refractory cohort, the measurement of strong efficacy, such as complete remissions, was not achieved [83]

Since ^223^Ra cannot be reliably chelated and then coupled to binding vector molecules, its parent nuclide thorium-227, with a half-life of 18.7 days, was used in a first human clinical trial for targeted alpha therapy with monoclonal antibodies against CD22 in lymphomas. In this study, the octadentate chelator 3,2-hydroxypyridinone (3,2-HOPO) was used. A small number of patients (*n* = 21) were treated with a maximum cumulative activity of 13.8 MBq, and complete remission (CR) was observed in one patient treated with 3.1 MBq and partial remission (PR) was observed in four patients treated with 1.5–4.6 MBq [84]. ^225^Ac was conjugated to the monoclonal antibody J591 targeting PSMA in prostate cancer is under investigation in a phase I clinical trial (ClinicalTrials.gov Identifier: NCT03276572).

However, in general, peptides as carrier molecules for alpha–emitting radionuclides are currently more commonly applied for solid tumors. The intravenous injection of longer half-lived radionuclides might reduce the background radiation of the carrier molecules show rapid renal elimination [85,86]. Consequently, the combination of the rapidly distributing somatostatin analogue with the short-lived radionuclide ^213^Bi was focused on intra-arterial application taking advantage of a high first-pass effect [87]. Patients that were refractory to beta-emitting radionuclides were treated with the cumulative activities of up to 20 GBq [^213^Bi]Bi–DOTA–TOC over several treatment cycles. During the follow-up, one patient (1/7) developed MDS and acceptable renal function impairment was observed. For the parent nuclide ^225^Ac, cumulative doses of up to 60–80 MBq [^225^Ac]Ac–DOTA–TOC resulted in acute hematologic toxicity with a platelet and leucocyte count nadir at 4–6 weeks with subsequent recovery. No hematological malignancies were observed, but renal toxicity was observed and two long-term survivors developed terminal kidney failure [88]. Based on the experience with ^225^Ac, clinical trials targeting the somatostatin receptor by [^225^Ac]Ac–DOTA–TOC are under way (ClinicalTrials.gov Identifier: NCT05477576) [89].

Furthermore, not only is [^225^Ac]Ac–DOTA–TOC under investigation, but there is also increasing experience with [^225^Ac]Ac–DOTA–TATE [90,91]. In a pilot study in metastatic paragangliomas, the [^225^Ac]Ac–DOTA–TATE ((100 kBq/kg) body weight per cycle at 8-weekly intervals up to a cumulative activity of ~ 74 MBq), in combination with the radiosensitizer capezetabine, achieved disease control in 8 of 9 patients treated [91].

In the future, it is expected that combined immuno- and endoradiotherapies will gain significance in personalized medicine, especially for established tumor mutations like BRCA, RET, and BRAF. A favorable outcome was recently reported in a patient with a G3 BRCA-mutated neuroendocrine tumor through a successful combination of olaparib and [^225^Ac]Ac–DOTA–TATE (see Figure 6). To further elucidate the effectiveness, safety, and efficacy of the aforementioned treatments, clinical trials are necessary.

In a clinical trial treating NET patients without prior peptide receptor radiotherapy with four cycles of 2.50 MBq/kg [^212^Pb]Pb–DOTAM–TATE, objective radiological responses were observed in most patients. However, diagnostic imaging was performed with [^68^Ga]Ga–DOTA–TATE, whereas [^203^Pb]Pb–DOTAM–TATE was used for human dose calculations [55]. Additionally, the Tyr^3^-octreotide (TOC) variant VMT-α-NET was used for human applications as a conjugate for ^203/212^Pb with high chelation properties [56]. SPECT/CT images (low dose, 224 MBq) with [^203^Pb]Pb–VMT-α-NET were acquired to assess the feasibility of [^212^Pb]Pb–VMT-α-NET therapy. A recent study demonstrated the post-treatment imaging of [^212^Pb]Pb–VMT-α-NET in a patient with metastatic NET [92]. The advantages of [^212^Pb]Pb–VMT-α-NET therapy are rapid renal clearance with potentially less nephrotoxicity than the standard radiopharmaceuticals and the possibility of dosimetry prediction using the elementally matched isotope ^203^Pb as an imaging surrogate (Figure 7).

With other solid tumors, target availability is typically a major challenge. Therefore, many attempts to introduce alpha particle-emitting radionuclides into clinical application followed locoregional applications where pharmacokinetics in the whole body are less relevant. A variety of applications have been proposed, for example, to treat the peritoneal cavity, the urinary bladder, or the intracerebral operation cavities after the resection of malignant cerebral tumors. Here, a broader choice of nuclides is possible because target-to-background is already high due to the route of application. ^211^At coupled to MX35-F(ab’)2 was applied to the peritoneal cavity for treating ovarian cancer with activities up to 355 MBq and no signs of radiation-induced toxicity were reported [93]. Also, ^212^Pb (τ½ = 10.6 days) coupled to the Her2/neu binding monoclonal antibody trastuzumab was considered safe for patients with peritoneal carcinomatosis up to 27 MBq/m^2^ [94]. ^213^Bi was proved to be safe in loco-regional treatment for in situ bladder carcinoma up to 821 MBq [95]. Other clinical approaches targeted residual tumor cells of high-grade brain tumors after operation. Here, the radionuclides ^213^Bi or ^225^Ac were labeled to substance-P and applied after surgical resection and considered as safe and well tolerated [96,97,98].

For systemic therapy, typically administered by i.v. route, the properties and availability of the molecular target are undoubtedly dominating factors for successful internal targeted endoradiotherapy. With the prostate-specific membrane antigen (PSMA), a highly expressed carboxypeptidase and a well available target on the cell surface is available and targeted alpha therapy is expected to contribute to further improvements in treating prostate carcinoma. Here, clinical data mainly from ^225^Ac were reported, but the radionuclide ^227^Th has also been suggested. ^227^Th transported by a monoclonal antibody targeting PSMA showed preclinical efficacy in several subcutaneous mouse models and is a candidate for clinical translation [99]. In thirteen patients treated with [^225^Ac]Ac-PSMA-617, the median overall survival was 8.5 months with a majority of patients showing prostate-specific antigen (PSA) responses [100]. By standardized quality of life questionnaires, a moderate improvement in the global health status was documented. One patient even reported exceeding a 5-year complete remission after [^225^Ac]Ac–PSMA-617 treatment [101]. In addition to PSMA-617, other PSMA-binding small molecules were also used as binding vectors in clinical application [102]. For example, [^225^Ac]Ac–PSMA–I&T applied in 1–5 cycles (6–8.5 MBq) was compassionately used in 18 patients, among which seven were experiencing a PSA response with the lowest PSA levels, which were < 50% to baseline. Despite being a well-suited target and although imaging has become increasingly applied in the therapeutic management of PSMA-targeted therapies, the imaging of ^225^Ac after therapy is—in contrast to the imaging of ^177^Lu—not well suited for diagnostic purposes. Low therapeutic activities and gamma ray emissions coming from the daughter nuclides preclude the quantification of targeting and post-therapeutic imaging might be mainly suited for quality control (Figure 8). Although the PSA response rates might be high in these patient cohorts, a number of patients did not show a response to PSMA-targeted alpha therapy. A molecular analysis of the non-responding tumor tissue with sufficient PSMA expression from seven patients after therapy with ^225^Ac-labeled PSMA-617 shows a high rate of alterations in the DNA damage-repair or cell cycle checkpoint [103]. This underlies the importance of the relationship between the tumor biology and targeted alpha therapy.

Taken together, advances in the development of therapeutic radiopharmaceuticals likely will trigger the vast expansion of clinical trials in the future. Clinical data from controlled, non-controlled trials, or clinical observations will be valuable to determine the extent of therapeutic effects in comparison, for example, to clinical experience and other reported clinical trials. Although prospective randomized trials are needed, such trials will be further complemented by, e.g., establishing broad safety profiles and dose–response relationships from compassionate use applications. In particular, with the aim of developing newly approved therapies, future randomized clinical trials might further elucidate the exact role of alpha emitters in patient care.

## 7. Outlook

One major field of application of α-radionuclide radiopharmaceuticals will be the clinical validation of several unique aspects that are described in the theoretical models and preclinical work. The exact impact of off-target and daughter redistribution on a clinical effect might depend on the pharmacokinetic details of the carrier, its individual variation, and the extent of internalization upon cell binding, and must be addressed in future work. In addition, the cytotoxic effects on tumors displaying a different biology and a different extent on biological variation are fields to be addressed for different tumor entities. Clinical trial design with regard to individual aspects like dosimetry including micro-dosimetric aspects will also remain challenging. Another major field will be the incorporation of imaging and a more sophisticated analysis like the parameters of heterogeneity and radiomics. In this regard, new tools such as artificial intelligence might contribute to both trial design and individualization by imaging guidance. Future developments will also need to address the role of alpha-emitting endoradiotherapy in several aspects where synergistic effects are expected. Several examples can be hypothesized like adjuvant systemic or locoregional treatment to maximize the advantage arising from the short high-LET pathways of alpha particles for single-tumor-cell diseases. Other highly promising scenarios include the combination of systemic alpha therapy with other approaches like chemotherapy or immune modulation therapy or the combination of internal with external radiation.

## Figures and Tables

**Figure 1 pharmaceuticals-17-00076-f001:**
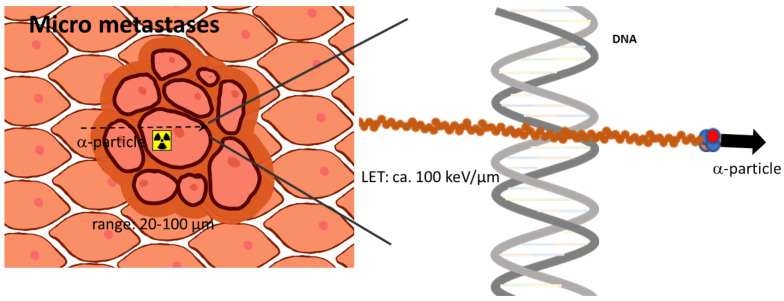
Scheme visualizing the prominent advantages of alpha-emitting radionuclides treating singles cells or a small tumor cell cluster. Due to the high linear energy transfer, alpha-induced DNA damage is higher than for other radiation like gamma or beta radiation.

**Figure 2 pharmaceuticals-17-00076-f002:**
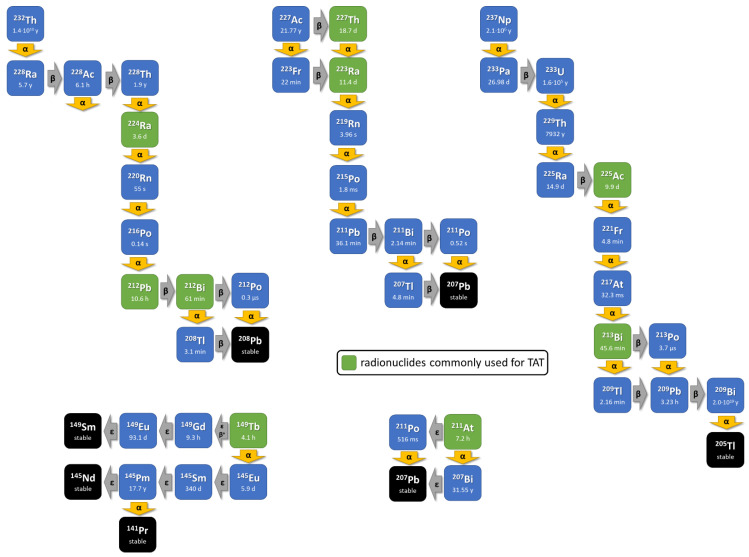
The principle of internal alpha radiation is enhanced with approaches that use radionuclides decaying via a short decay chain including further alpha-emitting isotopes. The decay schemes of the commonly used in vivo alpha radionuclides (green box).

**Figure 3 pharmaceuticals-17-00076-f003:**
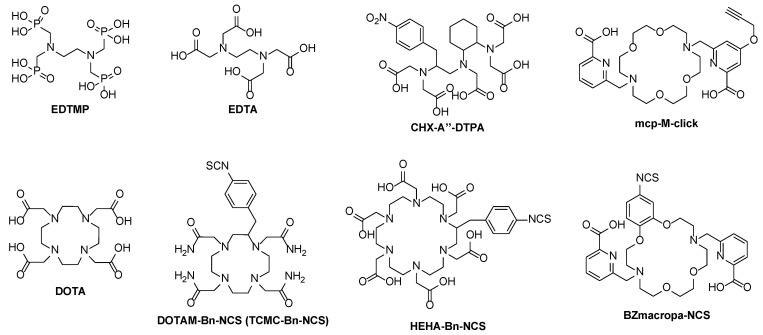
Examples of open-chain (EDTMP, EDTA, CHX-A″–DTPA) and macrocyclic chelators (DOTA, DOTAM–Bn-NCS, HEHA–Bn–NCS, BZmacropa–NCS) for ^225^Ac and its diagnostic radiometal matches.

**Figure 4 pharmaceuticals-17-00076-f004:**
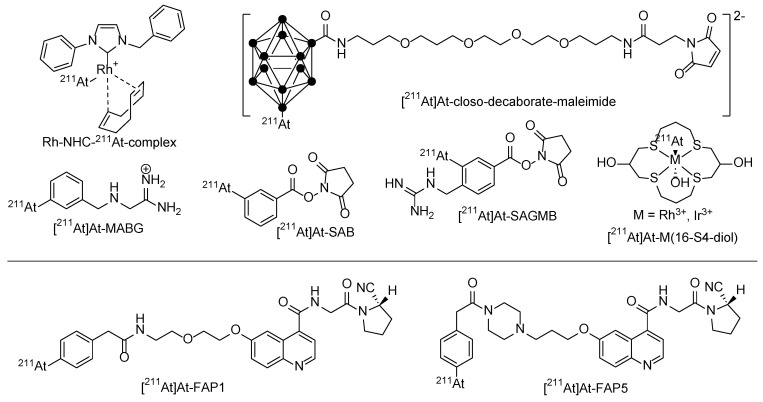
Different chemical strategies using prosthetic labeling groups to stably bind ^211^At for a later conjugation to the target molecule and two promising small-molecule ^211^At-radiotherapeutics [^211^At]At-FAPI1 and [^211^At]At-FAPI5.

**Figure 5 pharmaceuticals-17-00076-f005:**
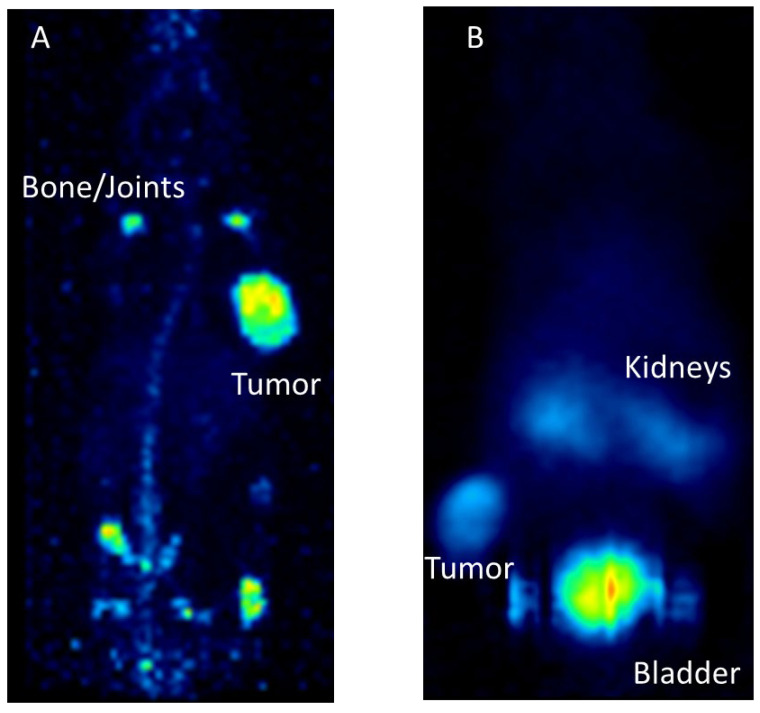
Preclinical examples for targeting with monoclonal antibodies or peptides. (**A**): [^89^Zr]Zr–Trastuzumab targeting Her2/neu in a mouse model carrying subcutaneous BT474 tumors. PET imaging was conducted 7 days after radiotracer application. (**B**): [^68^Ga]Ga–DOTA–TOC targeting the somatostatin receptor in a mouse model with subcutaneous AR42J tumor. PET imaging was performed one hour after tracer application.

**Figure 6 pharmaceuticals-17-00076-f006:**
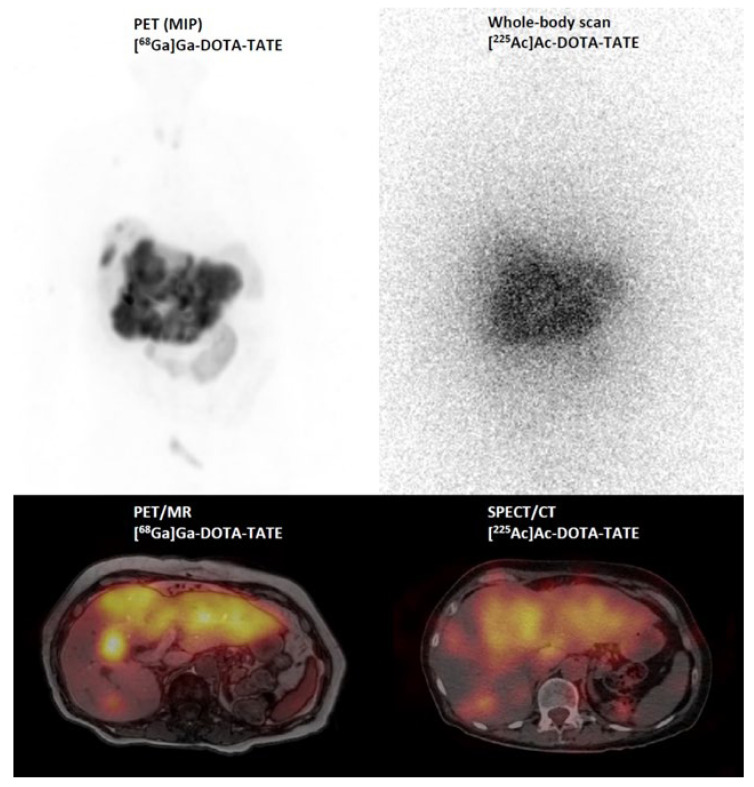
[^68^Ga]Ga–DOTA–TATE PET/MR (maximum intensity projection (MIP) and PET/MR fusion image) and post-therapy scintigraphic imaging (whole body and SPECT/CT fusion image) 24 h after injection of 6.5 MBq [^225^Ac]Ac–DOTA–DATE in a patient with a G3 neuroendocrine tumor.

**Figure 7 pharmaceuticals-17-00076-f007:**
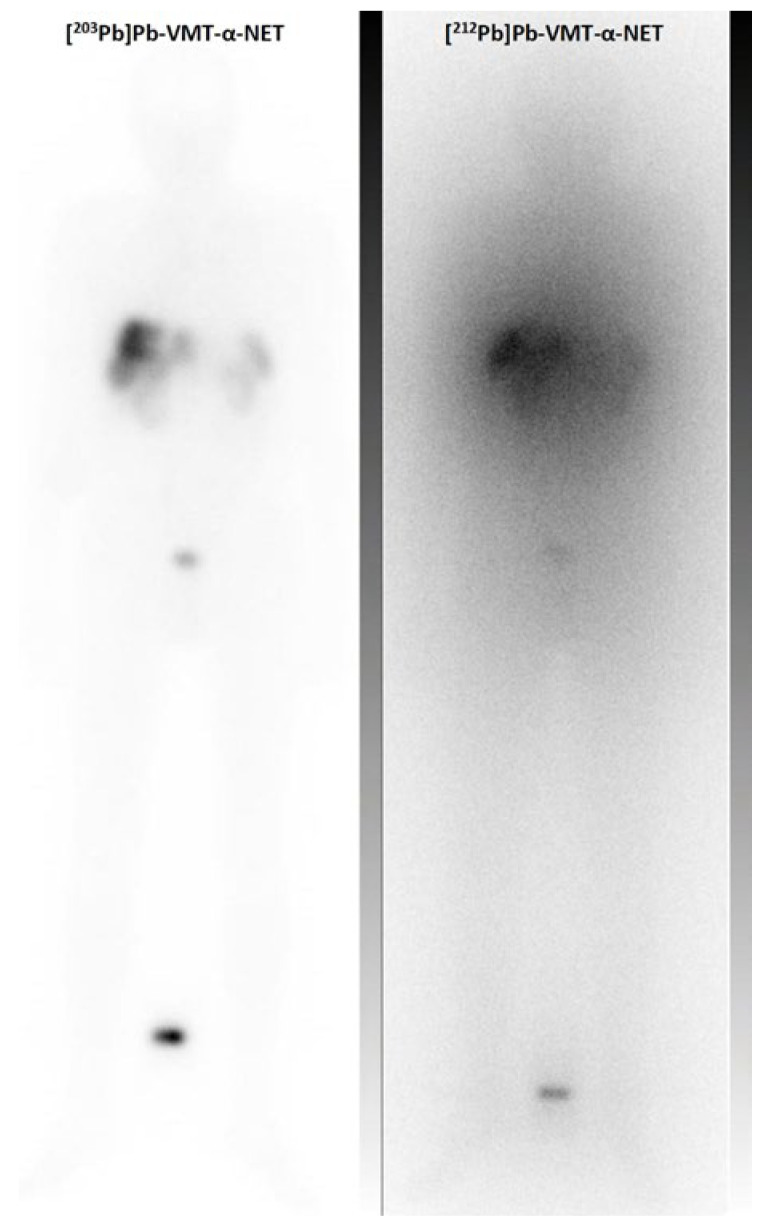
Clinical example of the scintigraphic imaging of the theranostic pair ^203^Pb and ^212^Pb: Pre-treatment whole body scan of [^203^Pb]Pb–VMT-α-NET (348 MBq, 2 h p.i.) and post-treatment imaging of [^212^Pb]Pb–VMT-α-NET (86 MBq, 2 h p.i.) in a patient with a G3 hepatic metastatic neuroendocrine tumor.

**Figure 8 pharmaceuticals-17-00076-f008:**
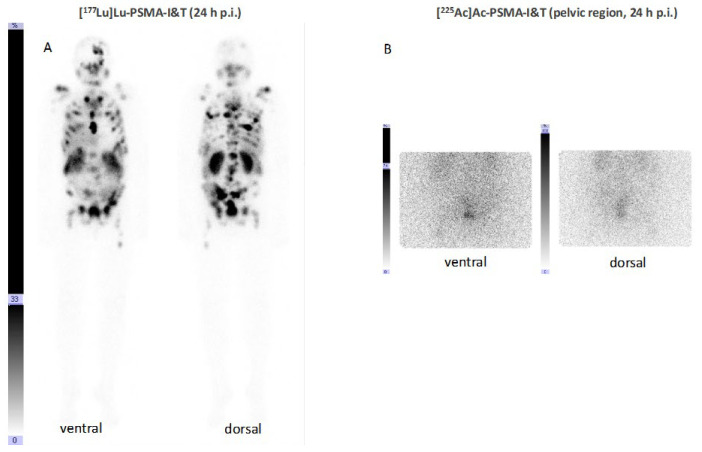
Anterior and posterior scintigraphic imaging of a patient treated with 5 GBq [^177^Lu]Lu–PSMA–I&T (**A**) and subsequently—due to rising PSA levels combined with reduced bone marrow reserve—with 8 MBq [^225^Ac]Ac–PSMA–I&T 6 months afterwards (**B**).

**Table 1 pharmaceuticals-17-00076-t001:** Chelating systems for ^225^Ac to pair with diagnostic radionuclides.

Chelator	Labeling Conditions	In Vivo Stability	Diagnostic RN
**EDTA/DTPA**	40 °C, 30 min	Failed	^68^Ga, ^43/44^Sc
**HEHA, PEPA, TETA, TETPA, and DOTPA**	95 °C, 60 min (HEHA) 40 °C, 30 min (PEPA)	Failed	^68^Ga, ^43/44^Sc, ^111^In
**DOTA**	90 °C, 30 min	Sufficient	^68^Ga, ^43/44^Sc, ^111^In, (^132/133^La)
**DO3APic**	25 °C, 30 min	Sufficient to low	^132/133^La
**Macropa**	rt, 5–15 min	High	^132/133^La

**Table 2 pharmaceuticals-17-00076-t002:** Comparison of bond energies of iodinated and astatine–alkyl and –aryl compounds.

	Alkyl Derivatives	Aryl Derivatives
**C–I bond energy**	220 kJ/mol	270 kJ/mol
**C–At bond energy**	160 kJ/mol	200 kJ/mol

## Data Availability

Data sharing not applicable.

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
