# Peer review of "Alpha-Emitting Radionuclides: Current Status and Future Perspectives"

_pharmaceuticals, 2024, doi:10.3390/ph17010076_

Round 1

Reviewer 1 Report

Comments and Suggestions for Authors

It's a very interesting and well-written paper on Alpha-emitting radionuclides.

I have several comments:

1. Figure 2 seems too complex, is there a way to simplify it or split it into several figures?

2. The PSMA-based RLT in prostate cancer is one of the most important applications, could you provide some illustrations? (after the figure 6). Otherwise, a figure from this publication ? Kratochwil C, Bruchertseifer F, Giesel FL, Weis M, Verburg FA, Mottaghy F, Kopka K, Apostolidis C, Haberkorn U, Morgenstern A. 225Ac-PSMA-617 for PSMA-Targeted α-Radiation Therapy of Metastatic Castration-Resistant Prostate Cancer. J Nucl Med. 2016 Dec;57(12):1941-1944.

3. I would suggest to have paragraphs on radiation protection and dosimetry aspects as they are also essential for this practice. These references could be helpful:

- Dauer, L.T., et al. Health Phys. 2014

- Thakral, P., et al. JNMT, 2020. 48(1): p. 68-72

- Chittenden, S.J., et al. JNM, 2015. 56(9)

- Pratt, B.E., et al. NMC, 2018. 39(2)

- Stabin, M.G. and J.A. Siegel. Health Phys, 2015. 109(3)

- Poeppel, T.D., et al. EJNMMI, 2018. 45(5)

- De Kruijff et al. Pharmaceuticals 2015; 8:321-336

- Kratochwil et al. J Nucl Med 2017;58(10):1624-1631 

Author Response

We want to thank both reviewers for their work and valuable comments and hope that we were able to address them sufficiently in the text and the point by point reply below.

Reviewer 1

Comments and Suggestions for Authors

It's a very interesting and well-written paper on Alpha-emitting radionuclides.

I have several comments:

  1. Figure 2 seems too complex, is there a way to simplify it or split it into several figures?

The decay schemes have been rearranged. However, we feel that there is not room for further simplification, since the decay schemes might be valuable as a readily accessible reference within the text

  1. The PSMA-based RLT in prostate cancer is one of the most important applications, could you provide some illustrations? (after the figure 6). Otherwise, a figure from this publication ? Kratochwil C, Bruchertseifer F, Giesel FL, Weis M, Verburg FA, Mottaghy F, Kopka K, Apostolidis C, Haberkorn U, Morgenstern A. 225Ac-PSMA-617 for PSMA-Targeted α-Radiation Therapy of Metastatic Castration-Resistant Prostate Cancer. J Nucl Med. 2016 Dec;57(12):1941-1944.

The following text and a new figure 8 on Ac-225 PSMA vs Lu-177 PSMA was introduced:

Despite being a well suited target and imaging has become increasingly applied in therapeutic management with PSMA targeted therapies, the imaging of 225Ac after therapy is - in contrast to imaging of 177Lu - not well suited for diagnostic purposes. Low therapeutic activities and gammy emission coming from daughter nuclides preclude quantification of targeting and post therapeutic imaging might be suited mainly for quality control (figure 8).

  1. I would suggest to have paragraphs on radiation protection and dosimetry aspects as they are also essential for this practice. These references could be helpful:

- Dauer, L.T., et al. Health Phys. 2014

- Thakral, P., et al. JNMT, 2020. 48(1): p. 68-72

- Chittenden, S.J., et al. JNM, 2015. 56(9)

- Pratt, B.E., et al. NMC, 2018. 39(2)

- Stabin, M.G. and J.A. Siegel. Health Phys, 2015. 109(3)

- Poeppel, T.D., et al. EJNMMI, 2018. 45(5)

- De Kruijff et al. Pharmaceuticals 2015; 8:321-336

- Kratochwil et al. J Nucl Med 2017;58(10):1624-1631 

Since the regulations on radiation protection are considerable heterogeneous and only minor scientific steps has been addressed in the recent years we have limited this topic to Radium-223 as the isotope with by far the most experience concerning radiation protection regulations. Therefore we have cited two important work that describe radiation safety aspects with Radium-223.

Reviewer 2 Report

Comments and Suggestions for Authors

In this manuscript, the authors report current status and perspectives of Alpha-emitting radionuclides. This manuscript is valuable for readers of Pharmaceuticals. It is recommended for publication with the following issues being addressed appropriately.

1)     In 4. Radioisotopes of lead for theranostics section, the authors are encouraged to provide the current production methods of 212Pb.

2)     In 6. Clinical overview and perspectives section, the authors are encouraged to summarize alpha radionuclide drugs that have entered the stage of clinical research in a Table.

3)     In the development of α-radionuclide radiopharmaceuticals, the acquisition of α-radionuclides (especially 225Ac and other nuclides), the off-target and daughter redistribution of α-radionuclides, and micro-scale dose estimation are all challenges. The authors are encouraged to add some discussion about this in outlook section.

4)     In the whole manuscript, In vivo should be in vivo.

5)     On page 14, Fig 7, 203Pb and 212Pb should be 203Pb and 212Pb.

6)     On page 14, Astatine-221 should be Astatine-211.

Comments on the Quality of English Language

OK

Author Response

We want to thank both reviewers for their work and valuable comments and hope that we were able to address them sufficiently in the text and the point by point reply below.

Reviewer 2

Comments and Suggestions for Authors

In this manuscript, the authors report current status and perspectives of Alpha-emitting radionuclides. This manuscript is valuable for readers of Pharmaceuticals. It is recommended for publication with the following issues being addressed appropriately.

1)     In 4. Radioisotopes of lead for theranostics section, the authors are encouraged to provide the current production methods of 212Pb.

The following sentence was added:

Different chromatographic generator systems were developed to isolate 212Pb based either on 228Th as mother nuclide or directly on 224Ra.

2)     In 6. Clinical overview and perspectives section, the authors are encouraged to summarize alpha radionuclide drugs that have entered the stage of clinical research in a Table.

This is an important point, but we feel that this would be difficult to address. The content of the paper has a focus also on clinical application and summarizes both compassionate use reports and prospectively planned clinical trials. Since compassionate use applications might depend on regional regulations, availability and also physicians’ experiences, to include all compassionate use applications would result in a rather confusing table. To restrict such a table to prospectively planned clinical trials would, however, give only a fraction of available information. Since we think that both aspects are addressed in the paper we refrain from including a summary table on clinical applications. To address the importance of compassionate use experience we have included another case example as figure 8.

3)     In the development of α-radionuclide radiopharmaceuticals, the acquisition of α-radionuclides (especially 225Ac and other nuclides), the off-target and daughter redistribution of α-radionuclides, and micro-scale dose estimation are all challenges. The authors are encouraged to add some discussion about this in outlook section.

The outlook section was expanded as follows:

One major field of application of α-radionuclide radiopharmaceuticals will be clinical validation of several unique aspects that are described in theoretical models and preclinical work. The exact impact of off-target and daughter redistribution on clinical effect might depend on pharmacokinetic details of the carrier, its individual variation and the extent of internalization upon cell binding and must be addressed in future work. In addition, the cytotoxic effects on tumors displaying different biology and different extent on biologic variation are fields to be addressed for different tumor entities. Clinical trial design in regard of individual aspects like dosimetry including micro-dosimetric aspects will also remain challenging. Another major field will be the incorporation of imaging and more sophisticated analysis like parameters of heterogeneity and radiomics. In this regard new tools such as artificial intelligence might contribute to both trial design and individualization by imaging guidance.

Also, some other sentences were clarified.

4)     In the whole manuscript, In vivo should be in vivo.

This was changed

5)     On page 14, Fig 7, 203Pb and 212Pb should be 203Pb and 212Pb.

This was changed

6)     On page 14, Astatine-221 should be Astatine-211.

This was changed

3)     In the development of α-radionuclide radiopharmaceuticals, the acquisition of α-radionuclides (especially 225Ac and other nuclides), the off-target and daughter redistribution of α-radionuclides, and micro-scale dose estimation are all challenges. The authors are encouraged to add some discussion about this in outlook section.

The outlook section was expanded as follows:

One major field of application of α-radionuclide radiopharmaceuticals will be clinical validation of several unique aspects that are described in theoretical models and pre-clinical work. The exact impact of off-target and daughter redistribution on clinical effect might depend on pharmacokinetic details of the carrier, its individual variation and the extent of internalization upon cell binding and must be addressed in future work. In addition, the cytotoxic effects on tumors displaying different biology and different extent on biologic variation are fields to be addressed for different tumor entities. Clinical trial design in regard of individual aspects like dosimetry including micro-dosimetric aspects will also remain challenging. Another major field will be the incorporation of imaging and more sophisticated analysis like parameters of heterogeneity and radiomics. In this regard new tools such as artificial intelligence might contribute to both trial design and individualization by imaging guidance.

4)     In the whole manuscript, In vivo should be in vivo.

This was changed

5)     On page 14, Fig 7, 203Pb and 212Pb should be 203Pb and 212Pb.

This was changed

6)     On page 14, Astatine-221 should be Astatine-211.

This was changed